# Assessment of the Quality of Mobile Applications (Apps) for Management of Low Back Pain Using the Mobile App Rating Scale (MARS)

**DOI:** 10.3390/ijerph17249209

**Published:** 2020-12-09

**Authors:** Adrian Escriche-Escuder, Irene De-Torres, Cristina Roldán-Jiménez, Jaime Martín-Martín, Antonio Muro-Culebras, Manuel González-Sánchez, Maria Ruiz-Muñoz, Fermín Mayoral-Cleries, Attila Biró, Wen Tang, Borjanka Nikolova, Alfredo Salvatore, Antonio I Cuesta-Vargas

**Affiliations:** 1Department of Physiotherapy, University of Malaga, 29071 Málaga, Spain; adrianescriche@gmail.com (A.E.-E.); cristina.roldan005@gmail.com (C.R.-J.); amuro@uma.es (A.M.-C.); mgsa23@uma.es (M.G.-S.); 2Instituto de Investigación Biomédica de Málaga (IBIMA), 29010 Málaga, Spain; detorres.irene@gmail.com (I.D.-T.); jaimemartinmartin@gmail.com (J.M.-M.); marumu@uma.es (M.R.-M.); fermin.mayoral.sspa@juntadeandalucia.es (F.M.-C.); 3Physical Medicine and Rehabilitation Unit, Hospital Regional Universitario, 29010 Málaga, Spain; 4Department of Human Anatomy, Legal Medicine and History of Science, University of Malaga, 29071 Málaga, Spain; 5Department of Nursing and Podiatry, University of Malaga, 29071 Málaga, Spain; 6Mental Health Unit, Regional Universitary Hospital of Malaga, 29010 Málaga, Spain; 7ITWare, 1117 Budapest, Hungary; attila.biro@itware.hu; 8Faculty of Science and Technology, Bournemouth University, Bournemouth BH12 5BB, UK; wtang@bournemouth.ac.uk; 9Arthaus, Production Trade and Service Company, 1000 Skopje, Macedonia; borjanka@arthaus.mk; 10Sensor ID Snc, 86021 Boiano, Italy; alfredo.salvatore@sensorid.it; 11Faculty of Health Science, School of Clinical Science, Queensland University Technology, Brisbane 4000, Australia

**Keywords:** application (app), exercise, mHealth, Mobile App Rating Scale (MARS), low back pain

## Abstract

Digital health interventions may improve different behaviours. However, the rapid proliferation of technological solutions often does not allow for a correct assessment of the quality of the tools. This study aims to review and assess the quality of the available mobile applications (apps) related to interventions for low back pain. Two reviewers search the official stores of Android (Play Store) and iOS (App Store) for localisation in Spain and the United Kingdom, in September 2019, searching for apps related to interventions for low back pain. Seventeen apps finally are included. The quality of the apps is measured using the Mobile App Rating Scale (MARS). The scores of each section and the final score of the apps are retrieved and the mean and standard deviation obtained. The average quality ranges between 2.83 and 4.57 (mean 3.82) on a scale from 1 (inadequate) to 5 (excellent). The best scores are found in functionality (4.7), followed by aesthetic content (mean 4.1). Information (2.93) and engagement (3.58) are the worst rated items. Apps generally have good overall quality, especially in terms of functionality and aesthetics. Engagement and information should be improved in most of the apps. Moreover, scientific evidence is necessary to support the use of applied health tools.

## 1. Introduction

Low back pain (LBP) is one of the most frequent types of pain [1] and the leading cause of disability worldwide, with a global prevalence of 7.2% according to the World Health Organization. LBP affects four in five individuals in their lifetime [2,3,4].

Current guidelines recommend non-pharmacological and non-invasive management, including advice to stay active, use of patient education, and exercise therapy [5,6]. Considering this, LBP care without medication is the preferred option for many clinicians [7,8,9]. Exercise is considered an effective tool to relieve pain and to improve the functional status of patients with chronic LBP [9,10,11]. Physically active interventions appear to have more potential to alter symptoms in chronic LBP than physically inactive interventions [12], being an inverse association between physical activity and LBP [13].

People with LBP express a strong desire for precise, consistent, personalised information on prognosis, treatment options, and self-management strategies related to healthcare and occupational issues [14]. Perceived fear-avoidance beliefs, catastrophic thinking, and perceived disability make the symptoms more severe than they are [15,16]. Cognitive-behavioural education may be used to increase physical performance, reduce pain level, and even decrease depression in people with LBP [17]. Exercise changes not only physical function but, also, other exercise-induced changes, such as improving psychological status and cognitions (e.g., reduced anxiety and fear). It has a significant influence on pain and disability [18,19].

A patient-centred approach is highly recommended to optimise the success rate of the behavioural change (i.e., becoming more active), tailoring the treatment to the preferences and attitudes of the individual. This approach includes continuous communication, education during all aspects of treatment, patient-defined goals, patient empowerment, self-monitoring and feedback [20,21,22].

Digital interventions may improve different behaviours and facilitate accessibility, minimising distance and time barriers [23]. Computer adaptation involves matching the intervention messages with the characteristics of the participants, based on responses to one or more assessments [24]. These messages could be used to impact persuasion and enhance the magnitude of the behavioural change [23,25,26,27,28]. 

However, the rapid proliferation of technological solutions often does not allow for the correct assessment of the quality of the tools [29,30]. Moreover, this assessment is often performed based on user opinions and satisfaction, or self-created scales that are not validated [31]. Most applications (apps) and digital health interventions applied, therefore, are not based on studies that support their effectiveness. It is necessary to carry out further studies to assess the effect of the tools currently available, as well as new ones to be developed. This fact will allow for evidence-based practice, ensuring a higher efficiency of intervention programmes.

The Mobile App Rating Scale (MARS) is an objective, valid, easy-to-use assessment tool designed for classifying and assessing the quality of mobile health (mHealth) apps [29]. This tool has shown to be a reliable [29,32] and widely applied [31,32,33,34,35,36,37,38,39,40,41,42,43] way to systematise assessment of the quality of mobile apps. However, training in the use of the tool and expertise in mHealth and the specific health field is required to administer it [44]. Accompanying appropriate training, the high level of interrater reliability obtained by this tool supports its use to allow health practitioners and researchers to use the scale with confidence [29]. The MARS includes 23 items grouped in different sections: engagement, functionality, aesthetics, information quality, and subjective quality [29]. 

It is suggested that an exhaustive analysis will allow practitioners and users to have relevant information for choosing an appropriate mHealth app. Thus, an exhaustive assessment of mobile apps available for interventions on LBP might be useful for clinicians, allowing them to know those most suited to encouraging patients with chronic LBP to become more active. The aim of this study, therefore, is to review which mobile apps are currently available in the app market, and to carry out an exhaustive assessment of their quality using the MARS, to provide an overview of their features, quality, and practicability.

## 2. Materials and Methods 

### 2.1. Information Sources and Search Strategy 

Two reviewers (physical therapists with experience in lower back pain (LBP) management and mobile health (mHealth), previously trained in the use of The Mobile App Rating Scale (MARS)) searched applications (apps) that included an intervention for LBP in the official stores of the two main operating systems, Android (Play Store) and iOS (App Store). The search was carried out in September 2019. To maximise the chances of recovering all the target results, a general term such as “low back pain” was used for the search. Since each platform offered different content depending on the region, the search process was repeated on both platforms for localisation in Spain and the United Kingdom.

### 2.2. Eligibility Criteria

Concerning this study, those applications (apps) related to lower back pain (LBP) that included feedback or intervention were selected. Apps that included other pain conditions with specific sections for LBP also were selected.

Apps based solely on general pain or that were in languages other than English or Spanish were excluded. Likewise, apps with significant technical problems and paid apps also were excluded.

Apps that were free to download and use but included exclusive paid content were not discarded, since they allowed the app to be used and evaluated. However, any free apps that subsequently required a subscription or that only allowed an initial assessment before requiring payment were excluded.

### 2.3. Application (App) Selection

The same independent reviewers screened the title and the download page of the found applications (apps). Potentially eligible apps were imported into a database (in case of doubt, they were imported), and duplicates from apps found in both regions were identified and unified. Remaining apps were downloaded, and any apps that did not meet the selection criteria were deleted. A third reviewer was consulted in the event of any doubts concerning the eligibility of an app.

### 2.4. Data Extraction and Quality Assessment

Two reviewers (physical therapists with experience in lower back pain (LBP) management and mobile health (mHealth), previously trained in the use of The Mobile App Rating Scale (MARS)) independently downloaded, used, and evaluated the remaining applications (apps), using an extraction form for data extraction. The form included data about the app developer, platform, version, year of publication of the last version, cost (paid or free app, and additional paid content), number of downloads, user rating, availability of privacy statement and privacy technical aspects, medical product information, and the items included in the MARS.

The MARS [29] was applied to assess the quality of the included apps. This scale was based on 23 items grouped in the following five sections:(A)Engagement (five items): Fun, interesting, customisable, interactive (e.g., sends alerts, messages, reminders, feedback, enables sharing), well-targeted to the audience.(B)Functionality (four items): App operation, easy to learn, navigation, flow logic, and gestural design of the app.(C)Aesthetics (three items): Graphic design, overall visual appeal, colour scheme, and stylistic consistency.(D)Information quality (seven items): Contained high-quality information (e.g., text, feedback, measurements, and references) from a credible source.(E)Subjective quality (four items): Personal interest in the app

Each item was scored from 1 (inadequate) to 5 (excellent), and a final score of app quality was obtained with the mean score of sections A, B, C, and D. The subjective app quality score was obtained independently with the average score of section E. Moreover, there were six app-specific items grouped in section F that assessed the perceived impact of the app on the user’s knowledge, attitudes, intentions of change, as well as the likelihood of an effective change toward the health behaviour in question. Due to the exclusion of the subjective quality section from the overall mean app quality score, MARS’s ability to assess quality objectively increased. Moreover, the high correlation between the MARS total score and the user’s star rating found in previous studies suggested that it was capturing the perceived overall quality adequately [29].

### 2.5. Data Synthesis and Analysis

Each reviewer independently assessed each of the Mobile App Rating Scale (MARS) items in each of the applications (apps). Regarding each item of each app, the mean value between the values of both reviewers was calculated. This mean value was used for the analysis of the mean of all the apps for each item and section of the MARS. Thus, average scores and standard deviations were retrieved from each section, as well as the total score and standard deviation of each app, allowing for a descriptive analysis. A third reviewer was consulted in the event of discrepancies in the descriptive items (items without a numerical score) between the two main reviewers. Following the example of a previous study [33], apps were classified into tertiles to facilitate the interpretation of the readers.

## 3. Results

A total of 500 applications (apps) were retrieved from the Play Stores of the United Kingdom and Spain, while 53 apps were found in the App Stores of both countries. After deleting duplicates and screening the title and the download page of the remaining apps, 34 Android and 15 iOS apps were selected as potentially eligible from the Play Store and App Store, respectively. After downloading and checking the fulfilment of the selection criteria, 17 apps finally were included in the descriptive analysis. Three of the included apps were obtained from the App Store, while the remaining 14 apps were retrieved from the Play Store. Table 1 and Table 2, respectively, show the details and the main characteristics of the included apps. Ten apps included a privacy statement detailing which information was collected and for what purpose (Table 2). Only six apps introduced login or password options to improve user’s data privacy (Table 2). One app (Healo) declared to be General Data Protection Regulation compliant and secure in accordance with the European Union regulation and to be registered with the Swedish Medical Products Agency (Table 2). Table 3 shows a summary description of how the apps work.

The mean score of the users of the ten apps that reported these data was 4.11 (Standard deviation (SD) 1.07) on a scale of 1–5 stars. According to the Mobile App Rating Scale (MARS) scoring, mean app quality was 3.81 (SD 0.43), ranging from 2.83 (the worst rated) to 4.57 (the most highly rated). Table 4 shows the score of each app according to the MARS.

## 4. Discussion

This study aimed to review which mobile applications (apps) are currently available in the app market and to carry out an exhaustive assessment of their quality using the Mobile App Rating Scale (MARS).

### 4.1. Quality

Described above in the Methods, the Mobile App Rating Scale (MARS) divides the application (app) scores into three dimensions. The assessment of sections A–D provides the mean quality score of the app. Subjective quality is scored according to the evaluation of section E. Finally, section F assesses the perceived impact of the app. This evaluation structure prioritises aspects such as engagement, functionality, aesthetics, and information over the subjective opinion of the assessor about the potential effectiveness of the app (E, subjective quality; F, perceived impact of the application). While digital health interventions require appropriate functional and design aspects, a high potential for impact and effectiveness are essential for continued use of the health tool. General assessment of the quality of the apps, therefore, should take into account the quality items assessed in sections A–D of the MARS, along with the score for subjective quality and app impact sections, as well as user rating. Moreover, not all aspects assessed on the scale have the same relevance for digital health interventions. Detailed analysis of the results in each section is essential in choosing an appropriate app.

It is difficult to deduce the characteristics that an ideal app should have. Obtaining high levels of engagement through appropriate strategies that motivate behaviour change is essential, on the one hand. However, engagement is significantly influenced by the functionality and ease of use of the app and aspects such as aesthetics that make it more attractive. It is essential, on the other hand, that the information provided by the app is of quality to determine if its use will be safe. The efficacy of the intervention using each specific mobile health (mHealth) app should be studied, but this rarely occurs. Legally, the handling of the patient’s private information will be a relevant aspect to know to comply with current regulations.

During this study, the average quality of the 17 included apps ranged between 2.83 (Right Motion—Alivia tu dolor solo con ejercicios) and 4.57 (Healthy Spine & Straight Posture—Back exercises/Columna vertebral sana & Postura recta) (mean 3.82) according to the MARS, scored on a scale from 1 (inadequate) to 5 (excellent). Considering sections A–D, the best scores were found in functionality (mean 4.7), followed by aesthetic content (mean 4.1). To contrast, information (mean 2.93) and engagement (mean 3.58) were the worst rated items. Regarding subjective quality (mean 2.6) and app-specific assessment (mean 2.82), where impact on the user is most relevant, the mean scores were low. However, in these sections, the variability between apps was significant, with scores ranging between 1.25 (6 Minute Back Pain Relief)–4.25 (Healo) in subjective quality assessment and 1.83 (Yoga Poses for Lower Back Pain Relief; Bella’s Lower Back Pain Exercises)–4.33 (Lower back yoga—floor class) in the specific items of the app, respectively. Overall, the mean score was good in most apps, with only one app (Right Motion—Alivia tu dolor solo con ejercicios) with a value below 3/5. However, the moderately low mean scores found in fundamental aspects such as engagement (3.58) and subjective assessment (2.6) should be analysed, due to the risk of developing technically well-designed applications that are unable to produce engagement and adequate adherence in users.

Substantial differences were found between the quality score obtained using the MARS and the score awarded by users using stars. Concerning the case of user rating, the score for the apps was higher (mean 4.01) than in the assessment using the MARS (mean 3.82), except in three of them. The MARS score is an objective tool based on technical criteria considered necessary for an appropriate application of the intervention. Conversely, user assessment is usually subjective and, although it does not refer to specific items, it is mostly influenced by the impact upon and the subjective appreciation of the user. This fact makes comprehensive analysis of both assessment methods essential, otherwise technically well-designed apps could be overrated while those with aspects that were better subjectively valued by users could be dismissed.

### 4.2. Tailored Interventions: Behaviour Change Techniques

One of the foundations of tailored interventions is the premise that changes in sociocognitive determinants (attitudes, beliefs of efficacy) favour behavioural change [45]. There are different hypotheses about the mechanisms that underlie this process, but most of them agree that it is necessary to transform intention into action [46]. Education or provision of information seems to be the main strategy to influence these determinants [45]. Consequently, an intervention that provides details about the benefits of a specific healthy behaviour could persuade a person to take actions oriented toward this type of behaviour, which in turn could lead to a major change in behaviour. Other strategies to favour this behavioural change could be the use of feedback, monitoring, reminders, or the creation of a social group. The Mobile App Rating Scale (MARS) has a list of theoretical background/strategies that may have been applied in applications to promote behaviour change. These strategies are based on evaluation, feedback, information and monitoring, the choice of goals, and the proposal of advice and training methods, among other behavior change techniques. Additionally, it has a list of possible technical aspects that enhance these strategies. The evaluator marks the strategies used by each app. During this study, information and education, followed by the advice, strategies, and skills training were the most used strategies to motivate behaviour change. The most widely used technical aspect to favour these strategies was sending reminders.

### 4.3. Lack of Evidence

Health interventions supported by mobile health (mHealth) applications (apps) have experienced a significant increase. However, this rapid proliferation often hinders appropriate assessment of the effectiveness and validity of the applied tools [30]. Consequently, although studying the feasibility, safety, and effectiveness of implementing a health intervention is essential, most of them are applied despite a lack of such evidence [47].

The Mobile App Rating Scale (MARS) assesses whether there is a scientific basis (if the app has been trialled/tested and verified by evidence published in scientific literature) supporting the use of apps through an item included in the information section [29]. No scientific evidence supporting use of any of the apps included in this study was found. These results are in-line with previous reviews carried out on mHealth apps focused on pain, in which a significant lack of scientific basis was found to support the use of the available apps [33,48]. Previously suggested by some authors, these results may be due to the commercial and non-scientific origin of the included apps [33]. It is possible that promoting the development of apps by academic and scientific institutions (and the collaboration with commercial partners) could help improve this aspect [33]. However, it seems that the long times required for research, with long delays with respect to the development time of new applications, could precisely be a possible cause for the lack of app evaluations [49]. Greater support of scientific and academic institutions with more resources to shorten the development and testing of this type of mHealth solution, therefore, could favour the proliferation of apps with evidence-based use.

### 4.4. Privacy and Confidentiality

Health applications (apps) handle users’ private information on health status, habits, or preferences. While any personal information must be handled with caution, health information is especially sensitive [50,51,52,53]. The law protects patients’ rights to the confidentiality of their health data, but such laws are usually drafted for application in the health system. Applying and monitoring compliance with these rights in mobile health (mHealth) apps is, therefore, a challenge [51]. The Mobile App Rating Scale (MARS) includes items about whether access controls have been introduced in the app using login or password options to improve privacy. However, these items are included in a section of technical aspects of the apps; this is merely for descriptive purposes and does not influence the quality score. During this study, six apps (Back pain relief exercises; Regimen- back pain relief; Healo; Bella’s Lower Back Pain Exercises; Healure: Physiotherapy Exercise Plans; Curable: Back Pain, Migraine & Chronic Pain Relief) introduced login and password options. This may be essential as a first step to maintaining the confidentiality and security of user data. However, this information does not cover what happens with these data after collection. Storage and possible use of personal data by the app developers or third-party use are other fundamental aspects to analyse in the preservation of privacy and confidentiality. However, in the case of apps, the user does not usually have access to this information. A previous study found that most apps do not follow well-known practices and guidelines, threatening user privacy [50]. This circumstance could be improved if the developers informed users with privacy policy statements about possible uses of the user’s information, but this statement is often absent [54]. Concerning our study, ten apps (Back pain relief exercises; Regimen- back pain relief; Yoga Poses for Lower Back Pain Relief; Lower Back Pain Exercises; WomenBackWorkout; Healthy Spine & Straight Posture–Back exercises/Columna vertebral sana & Postura recta; Healo; Bella’s Lower Back Pain Exercises; Healure: Physiotherapy Exercise Plans; Curable: Back Pain, Migraine & Chronic Pain Relief) included a privacy statement detailing which information is collected and for what purpose. One app (Healo) declared to be General Data Protection Regulation compliant in accordance with European Union regulation.

### 4.5. Safety

Safety in the use of applications (apps) that can be used for health purposes is an essential aspect for practitioners. Clear indications about those activities and exercises that can be performed autonomously at home, or those that should be performed supervised, must be indicated for an adequate prescription and use. Information about possible limits related to the health status of users is also essential. Some of them had an initial profile that allows the intervention to be personalised. However, the adequacy of this assessment is also unclear. Some of the apps included disclaimers, stating that the services offered are for informational purposes only and not for professional medical advice. Thus, five apps (6 Minute Back Pain Relief; Back Doctor (FREE) Health. Stretch. Workout; Curable: Back Pain, Migraine & Chronic Pain Relief; Yoga Poses for Lower Back Pain Relief; Escuela de Espalda) declared that the information provided by the app is not a substitute for a medical service, recommending visiting the doctor before starting the programme. Only one app, (Healo), declares to be secure in accordance with European Union regulation and to be registered as a medical product (Swedish Medical Products Agency). Developers should consider this aspect in future apps.

The Mobile App Rating Scale (MARS) evaluates the information provided by the developers and the credibility of the source (conflicts of interest, commercial interest, academic origin, etc.). Additionally, MARS describes the evidence base of the apps (if the app has been trialled/tested and verified by evidence published in scientific literature). None of the included apps was found to be supported by evidence published in the scientific literature. This fact may be related to the commercial origin of all the included apps. Regarding this, the long times required for research, with long delays compared to the development time of new applications, have been attributed as a possible cause for the lack of app evaluation [49].

### 4.6. Recommended Applications (Apps)

“Healo” is the only application (app) that declares to be registered as a medical product. Obtaining an average quality score of 4.2 (1.23), it is one of the best-rated apps in this study. It had high scores in the engagement, functionality, and aesthetics sections. Additionally, the subjective quality score and the perceived impact of the app on the user are high. However, the information provided in the app description is insufficient. The advanced self-diagnosis tools it contains, the availability of a chat with health care advisers, the adaptation of the programmes to each user, as well as its attractive design, make it a potentially useful mobile health (mHealth) app.

“Healthy Spine & Straight Posture—Back exercises/Columna vertebral sana & Postura recta“ is the app with the highest objective score using the The Mobile App Rating Scale (MARS) scale [4.57 (0.63)]. It is a well-configured functional app that allows one to carry out exercise programmes in a guided way. The app allows the user to self-assess and track the progress of the programme and the results of the assessments. However, this app requires payment to have full access to the contents.

“Curable: Back Pain, Migraine & Chronic Pain Relief” is the only one of the included apps that does not offer an exercise-based intervention. Focused on patient education and pain neuroscience, it uses an attractive design through a narrative in the form of conversation, with text and audio. Obtaining an average objective quality score of 4.38 (0.82), it is one of the best-rated apps. It had high scores in the engagement, functionality, and aesthetics sections. Additionally, this app obtained high scores in the quality of the information and in the subjective perception of the quality and its impact on the user.

### 4.7. Study Limitations

The main limitation of this review is the exclusion of paid applications (apps). The search was carried out in duplicate in app stores in Spain and the United Kingdom. Therefore, apps available only in stores of other countries could not be retrieved. The app market is constantly updating. Although this may be a limitation, this review faithfully shows the current state of the market at the time of evaluation.

## 5. Conclusions

This study offers an analysis and description of applications (apps) available for managing lower back pain (LBP), to help practitioners and users to recommend quality mobile health (mHealth) apps adapted to the needs of each patient. The assessed apps generally had good overall quality, especially in terms of functionality and aesthetics. However, some apps must improve aspects such as engagement and information to increase their impact on users and ensure better security and privacy. “Healo” and “Healthy Spine & Straight Posture—Back exercises/Columna vertebral sana & Postura recta“ are mHealth apps with high objective and subjective quality scores that include an exercise-based intervention. “Curable: Back Pain, Migraine & Chronic Pain Relief” is another well-rated app that includes a neuroscience pain-based intervention programme. Moreover, additional scientific evidence is necessary to support the use of each mHealth app. Therefore, despite the extensive volume of apps on the market, this review shows an absence of validity studies, studies of psychometric characteristics, and clinical trials that demonstrate the effectiveness of these apps in people with LBP. This indicates the need to develop studies to allow the use of validated, effective apps supported by evidence. The safety of use and privacy also should be improved in most of the apps.

## Figures and Tables

**Table 1 ijerph-17-09209-t001:** Description of the lower back pain (LBP)-related applications (apps) included in the study.

App name	Platform	Paid Content	Version	Year (Last Version)	Downloads	Developer	Affiliations
Back pain relief exercises	App Store	Yes	2.5.6	2015	N/A	IREHAB.com	Commercial
Lower back yoga—floor class	App Store	Yes	2.5.0	2015	N/A	Centre de Yoga—La Source SARL	Commercial
Regimen- back pain relief	App Store	No	1.02	2018	N/A	Oyebimpe Oguntola	Commercial
6 Minute Back Pain Relief	Play Store UK/Spain	No	3.7	2019	100,000+	Round1Fight	Commercial
Yoga Poses for Lower Back Pain Relief	Play Store UK/Spain	No	2.2	2018	10,000+	Gonga Dev	Commercial
Lower Back Pain Exercises	Play Store UK/Spain	No	1.0	2019	1000+	1Bestofall	Commercial
Escuela de Espalda	Play Store UK/Spain	No	1.1	2019	5000+	Alberto Sanchez	Commercial
WomenBackWorkout	Play Store UK/Spain	No	1.0	2019	1000+	1Bestofall	Commercial
Healthy Spine & Straight Posture—Back exercises/Columna vertebral sana & Postura recta	Play Store UK/Spain	Yes	3.3	2019	1,000,000+	mEL Studio	Commercial
Back Doctor (FREE) Health, Stretch, Workout	Play Store UK/Spain	No	1.3	2017	10,000+	Dr. Watkins	Commercial
Right Motion—Alivia tu dolor solo con ejercicios	Play Store Spain	Yes	3.4	2019	5000+	Thiago Nishida	Commercial
Healo	Play Store UK/Spain	No	1.0.10	2019	100+	Empowered Applications	Commercial
Doado, Your Back Companion	Play Store Spain	No	2.0.2	2018	10,000+	Doado	Commercial
Bella’s Lower Back Pain Exercises	Play Store UK/Spain	Yes	1.0.1	2019	10+	PM Health	Commercial
Healure: Physiotherapy Exercise Plans	Play Store UK/Spain	Yes	1.1.07	2019	10,000+	Healure Technology	Commercial
Curable: Back Pain, Migraine & Chronic Pain Relief	Play Store UK/Spain	Yes	4.1.0	2018	10,000+	Curable Inc.	Commercial
Improve Posture For A Healthy Spine	Play Store UK/Spain	Yes	1.1	2019	1000+	Stay Fit With Samantha	Commercial

**Table 2 ijerph-17-09209-t002:** Characteristics of the included lower back pain (LBP)-related applications (apps).

App Name	Focus (What the App Targets)	Theoretical Background/Strategies	Technical Aspects of App	Privacy Data Technical Aspects	Privacy Statement	Medical Product
Back pain relief exercises	Physical health	Information/education; Advice tips/Strategies/Skills Training	Sends reminders	L; P	Yes	No
Lower back yoga—floor class	Physical health	Information/education; Advice tips/Strategies/Skills Training	Sends reminders	No	No	No
Regimen- back pain relief	Physical health	Information/education; Advice tips/Strategies/Skills Training	Sends reminders	L; P	Yes	No
6 Minute Back Pain Relief	Physical health	Information/education; Feedback; Advice/Tips/Strategies/Skills training	Sends reminders; Allows sharing (e.g., Facebook, Twitter)	No	No	No
Yoga Poses for Lower Back Pain Relief	Physical health	Information/education; Mindfulness/Meditation; Relaxation; Advice tips/Strategies/Skills Training	Sends reminders	No	Yes	No
Lower Back Pain Exercises	Physical health	Information/education; Advice tips/Strategies/Skills Training	Sends reminders	No	Yes	No
Escuela de Espalda	Physical health	Information/education; Feedback; Assessment; Advice tips/Strategies/Skills Training		No	No	No
WomenBackWorkout	Physical health	Information/education; Advice tips/Strategies/Skills Training	Sends reminders	No	Yes	No
Healthy Spine & Straight Posture—Back exercises/Columna vertebral sana & Postura recta	Physical health	Information/education; Feedback; Assessment; Advice tips/Strategies/Skills Training	Sends reminders	No	Yes	No
Back Doctor (FREE) Health, Stretch, Workout	Physical health	Information/education; Feedback; Advice tips/Strategies/Skills Training		No	No	No
Right Motion—Alivia tu dolor solo con ejercicios	Physical health	Information/education; Feedback; Advice tips/Strategies/Skills Training	Sends reminders	No		No
Healo	Physical health	Information/education; Feedback; Assessment; Advice tips/Strategies/Skills Training	Sends reminders	L; P	Yes	Yes
Doado, Your Back Companion	Physical health	Information/education; Feedback; Assessment; Advice tips/Strategies/Skills Training		No	No	No
Bella’s Lower Back Pain Exercises	Physical health	Information/education; Feedback; Advice tips/Strategies/Skills Training		L; P	Yes	No
Healure: Physiotherapy Exercise Plans	Physical health	Information/education; Feedback; Assessment; Advice tips/Strategies/Skills Training	Sends reminders	L; P	Yes	No
Curable: Back Pain, Migraine & Chronic Pain Relief	Physical health	Information/education; Feedback; Assessment; Advice tips/Strategies/Skills Training		L; P	Yes	No
Improve Posture For A Healthy Spine	Physical health	Information/education Feedback; Advice tips/Strategies/Skills Training		No	No	No

L = log-in; P = password App = application.

**Table 3 ijerph-17-09209-t003:** Summary description of how the applications (apps) work.

App Name	How the Apps Work
Back pain relief exercises	The user chooses their pain area and receives a list of suggested exercises. There is a chat to contact health care advisers. It contains premium exercises unlocked by monthly payment. The app allows the user to set reminders.
Lower back yoga–floor class	The application (app) offers examples of exercises and yoga poses to be performed, as well as tips for back care. Free access to sample exercises. The full content is available under subscription for three months. It allows the user to set reminders.
Regimen- back pain relief	The app offers examples of guided exercises depending on the objective selected by the user. It allows the user to set reminders.
6 Minute Back Pain Relief	The app offers nine guided exercises related to yoga poses. It allows the user to set reminders.
Yoga Poses for Lower Back Pain Relief	The app offers information related to back pain and guided exercises related to yoga poses. It allows the user to set reminders.
Lower Back Pain Exercises	The app offers guided exercises. It allows the user to configure the difficulty of the exercises and set reminders.
Escuela de Espalda	The app includes extensive information related to anatomy, biomechanics, exercise, postural hygiene, lifestyles, diagnostic tests, and pain management strategies. It provides evaluation tests and questionnaires to be completed by the user that allows follow-up, as well as guided exercises. User can select favourite exercises. The app includes a section with utilities for the healthcare professional.
WomenBackWorkout	The app offers guided exercises. It allows the user to configure the difficulty of the exercises and set reminders.
Healthy Spine & Straight Posture–Back exercises/Columna vertebral sana & Postura recta	The app contains some exercise programmes to be carried out in a guided way, as well as a battery of explained exercises. The user can self-evaluate. It allows the user to set reminders and configure the training frequency. Requires payment for full access to programmes.
Back Doctor (FREE) Health. Stretch. Workout	The app includes exercise programmes for neck and back pain, as well as chronic pain and other conditions. It provides information about lifestyles.
Right Motion–Alivia tu dolor solo con ejercicios	The app offers guided exercises. It allows the user to set reminders.
Healo	The app includes exercise programmes with exercises described in text and video. Exercise programmes are personalised depending on the results of the diagnosis received by a doctor or through a self-assessment in the app. The app allows guided self-diagnostics that include the location (body chart) and the characteristics of the pain, its onset, or its duration, among others. It asks the user questions trying to find red flags. The app offers you a potential diagnosis and an associated exercise programme in which reminders can be set. It includes a chat with health care advisers. The design and graphs of the app are very attractive.
Doado. Your Back Companion	The app includes explanatory videos of exercises and recommended postures in various situations of daily life. It allows the user to evaluate their pain to keep track of it.
Bella’s Lower Back Pain Exercises	The app includes an exercise programme divided into progressive phases. The exercises are explained in writing, as well as in audio and video. It requires payment for access to full content.
Healure: Physiotherapy Exercise Plans	The app contains some exercise programmes to be carried out in a guided way, as well as assessments related to the pain and symptoms. It allows the user to schedule programme sessions and set reminders. It requires payment for access to full content.
Curable: Back Pain, Migraine & Chronic Pain Relief	The app offers a narrative in the form of a written and listened conversation through which it discovers information related to pain. The intervention is focused on the neuroscience of pain and patient education. The design of the app is very attractive.
Improve Posture For A Healthy Spine	The app contains some exercise programmes to be carried out in a guided way, as well as a battery of explained exercises. It allows user to set reminders. It requires payment for access to full content.

**Table 4 ijerph-17-09209-t004:** Mobile App Rating Scale (MARS) scoring of the lower back pain (LBP)-related applications (apps).

App Name	App Quality Mean (SD)	Ta	Section Mean (SD)						User’s Stars Score (Votes)
			A	B	C	D	E	F	
Back pain relief exercises	4.11 (0.70)	T1	3.6 (0.89)	5 (0)	4.33 (0.58)	3.5 (1.97)	3 (1.41)	4.17 (0.98)	1.0 (5)
Lower back yoga—floor class	4.11 (0.70)	T1	3.6 (1.14)	5 (0)	4.33 (0.58)	3.5 (1.97)	3.25 (1.5)	4.33 (1.03)	N/A
Regimen- back pain relief	3.8 (0.30)	T2	3.6 (0.89)	4.25 (0.5)	3.67 (0.58)	3.67 (1.5)	2.75 (1)	3 (0)	N/A
6 Minute Back Pain Relief	3.99 (0.76)	T1	3.8 (0.84)	5 (0)	4 (0)	3.17 (1.33)	1.25 (1.26)	2.16 (1.33)	4.4 (1301)
Yoga Poses for Lower Back Pain Relief	3.17 (1.41)	T3	3.4 (1.14)	5 (0)	1.67 (0.58)	2.6 (1.14)	2.25 (0.5)	1.83 (0.98)	4.5 (326)
Lower Back Pain Exercises	3.82 (0.89)	T2	3.8 (0.84)	5 (0)	3.67 (0.58)	2.83 (0.98)	2.25 (0.96)	2 (1.09)	4 (6)
Escuela de Espalda	3.91 (1.02)	T2	2.8 (0.84)	4.5 (0.58)	5 (0)	3.33 (1.21)	2.25 (0.96)	2.67 (1.21)	5 (28)
WomenBackWorkout	3.82 (0.89)	T2	3.8 (0.84)	5 (0)	3.67 (0.58)	2.83 (0.98)	3.5 (0.96)	2 (1.09)	4 (6)
Healthy Spine & Straight PostureBack exercises/Columna vertebral sana & Postura recta	4.57 (0.63)	T1	4.6 (0.55)	5 (0)	5 (0)	3.67 (1.50)	2.25 (1.73)	3.33 (1.63)	4.6 (6246)
Back Doctor (FREE) Health. Stretch. Workout	3.43 (1.04)	T3	3 (1)	4.25 (0.96)	4.33 (0.58)	2.17 (0.75)	1.5 (0.96)	2.83 (0.98)	4.4 (78)
Right Motion—Alivia tu dolor solo con ejercicios	2.83 (0.79)	T3	2.4 (0.89)	4 (0)	2.33 (0.58)	2.6 (0.89)	1.75 (1.5)	2 (0.63)	4.4 (104)
Healo	4.2 (1.23)	T1	4.4 (0.55)	5 (0)	5 (0)	2.4 (0.89)	4.25 (0.58)	3.5 (0.84)	4.2 (5)
Doado. Your Back Companion	3.75 (0.97)	T3	3 (1.22)	4.5 (0.58)	4.67 (0.58)	2.83 (0.98)	2.75 (0.5)	2.83 (0.75)	4.6 (498)
Bella’s Lower Back Pain Exercises	3.8 (0.89)	T3	2.8 (1.3)	4.75 (0.5)	4.33 (0.58)	3.33 (1.21)	1.75 (0.5)	1.83 (0.41)	N/A
Healure: Physiotherapy Exercise Plans	3.47 (0.86)	T3	4 (0.70)	4 (0)	3.67 (0.58)	2.2 (0.84)	2.75 (1.26)	2.83 (1.16)	3.8 (155)
Curable: Back Pain, Migraine & Chronic Pain Relief	4.38 (0.82)	T1	4.6 (0.55)	4.75 (0.5)	5 (0)	3.17 (1.17)	4 (1.41)	4 (0)	3.8 (159)
Improve Posture For A Healthy Spine	3.9 (1.43)	T2	3.6 (1.14)	5 (0)	5 (0)	2 (0.7)	2.75 (1.26)	2.67 (1.03)	3.5 (8)
TOTAL MEAN	3.82		3.58	4.7	4.1	2.93	2.6	2.82	4.01

T: tertile; Tertile legend: T1: best-rated applications (apps); T2: average apps; T3: worst-rated apps; A: Engagement; B: Functionality; C: Aesthetics; D: Information; E: App subjective quality; F: App specific; N/A: not applicable.

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
