# Peer review of "Assessment of the Quality of Mobile Applications (Apps) for Management of Low Back Pain Using the Mobile App Rating Scale (MARS)"

_ijerph, 2020, doi:10.3390/ijerph17249209_

Round 1

Reviewer 1 Report

This article could concern a topic of interest in the public health fields, specially now that we are living with SARS-CoV-2 pandemic. To able to access online resources for maintaining well-being is a great opportunity, and it is useful, even for physicians and health experts, to know which tools to report to their patients. However, some observations are needed.

  • First, I have some concerns about the MARS scale, i.e. on its application by just two auditors and the method subjectivity. What reproducibility does this approach have? Are the reviewers some researchers or users with low back pain? Please better explain it.
  • Has safety in use been assessed, overall or for each app? These are exercises or indications that can always be done at home or limits of use are indicated (e.g. people with herniated discs or fractures). I believe this information is essential to doctors. If it is not possible to have this information, I think it is necessary to talk about it
  • It would be useful to understand if from this review it is possible to deduce the characteristics that an ideal app should have to provide not only points of reference to users but also feedback and ideas to producers. If possible, I suggest making these considerations in the discussion.
  • Not including paid information is a limit that should be exceeded if possible.
  • Data synthesis and analysis should be improved.
  • The authors should do a general linguistic check and correct some minor errors.

Author Response

Response to Reviewer 1 Comments

Comments:

This article could concern a topic of interest in the public health fields, specially now that we are living with SARS-CoV-2 pandemic. To able to access online resources for maintaining well-being is a great opportunity, and it is useful, even for physicians and health experts, to know which tools to report to their patients

Author’s response: We sincerely thank your positive feedback.

However, some observations are needed. First, I have some concerns about the MARS scale, i.e. on its application by just two auditors and the method subjectivity. What reproducibility does this approach have? Are the reviewers some researchers or users with low back pain? Please better explain it.

Author’s response: The MARS has demonstrated high levels of interrater reliability for evaluating the quality of mHealth apps on well-being [1] and mindfulness [2]. In fact, it is a tool designed for use by researchers and health practitioners. Although it is an easy-to-use tool, the standard professional version requires training and expertise in mHealth and the health field being studied to administer it. In this study, both reviewers were physical therapists with experience in low back pain management and mHealth field, previously trained in the use of MARS. There is also a version of MARS for app users (uMARS)[3], but in this study, the standard version for professionals has been used. This information has been included in the manuscript.

Currently, there are dozens of articles in the scientific literature using this approach for evaluating the quality of mHealth apps using MARS in pain [4], behavior change [5], skin cancer [6], gestational diabetes [7], rheumatology [8], quality of deaf and hard-of-hearing [9], COVID-19 [10], postpartum depression [11], mindfulness [2], weigth management [12], oral hygiene apps [13], sleep self-management [14], or potential drug-drug interactions [15], among many others.

Has safety in use been assessed, overall or for each app? These are exercises or indications that can always be done at home or limits of use are indicated (e.g. people with herniated discs or fractures). I believe this information is essential to doctors. If it is not possible to have this information, I think it is necessary to talk about it

Author’s response: Thank you for your comment. Safety in the use of apps that can be used for health purposes is an essential aspect for practitioners. Clear indications about those activities and exercises that can be performed autonomously at home or those that should be performed supervised must be indicated for an adequate prescription and use. Information about possible limits related to the health status of users are also essential. Some of the apps included in this study included an initial profile that allows the intervention to be personalized. However, the adequacy of this assessment is also unclear.

Some studies declared that the information provided by the app is not a substitute for a medical service, recommending visiting the doctor before starting the program. This may call into question the safety or efficacy of your interventions. Only one app (Healo) declares to be secure in accordance with European Union regulation, and to be registered as a medical product.

A new section has been included in the discussion of the manuscript dealing with this important topic.

It would be useful to understand if from this review it is possible to deduce the characteristics that an ideal app should have to provide not only points of reference to users but also feedback and ideas to producers. If possible, I suggest making these considerations in the discussion.

Author’s response: Thanks for your suggestion. We have introduced a small paragraph in the discussion with the characteristics of an ideal app, as well as a brief description of those recommended apps according to the results of this study.

Not including paid information is a limit that should be exceeded if possible.

Author’s response: Thank you for your comment. We agree that not including paid applications is a limit to establish which are the best applications in a specific field. However, the greater possibility of access of the free apps usually means that they have a greater scope, and for this reason they are more often selected by health practitioners to increase the possibilities of use by patients, whatever their economic situation.

Data synthesis and analysis should be improved.

Author’s response: Thank you for your recommendation. Similar sources and articles have been consulted trying to identify synthesis and analysis methods that could improve the present study. Relevant information about security, privacy and information about the registration as a medical product has been included in Table 2. The description of the methodology followed to calculate the data in Table 3 has been improved. A new Table (Table 4) describing how the apps work reinforces the information obtained from the apps, providing additional information to that provided by the numerical values.

The authors should do a general linguistic check and correct some minor errors.

Author’s response: Thanks for your recommendation. A linguistic check has been carried out to improve this aspect.

References

  1. Stoyanov, S.R.; Hides, L.; Kavanagh, D.J.; Zelenko, O.; Tjondronegoro, D.; Mani, M. Mobile App Rating Scale: A New Tool for Assessing the Quality of Health Mobile Apps. JMIR mHealth and uHealth 2015, 3, e27, doi:10.2196/mhealth.3422.
  2. Mani, M.; Kavanagh, D.J.; Hides, L.; Stoyanov, S.R. Review and Evaluation of Mindfulness-Based iPhone Apps. JMIR Mhealth Uhealth 2015, 3, e82, doi:10.2196/mhealth.4328.
  3. Stoyanov, S.R.; Hides, L.; Kavanagh, D.J.; Wilson, H. Development and Validation of the User Version of the Mobile Application Rating Scale (uMARS). JMIR Mhealth Uhealth 2016, 4, e72, doi:10.2196/mhealth.5849.
  4. Salazar, A.; de Sola, H.; Failde, I.; Moral-Munoz, J.A. Measuring the Quality of Mobile Apps for the Management of Pain: Systematic Search and Evaluation Using the Mobile App Rating Scale. JMIR Mhealth Uhealth 2018, 6, doi:10.2196/10718.
  5. McKay, F.H.; Wright, A.; Shill, J.; Stephens, H.; Uccellini, M. Using Health and Well-Being Apps for Behavior Change: A Systematic Search and Rating of Apps. JMIR Mhealth Uhealth 2019, 7, e11926, doi:10.2196/11926.
  6. Steeb, T.; Wessely, A.; French, L.E.; Heppt, M.V.; Berking, C. Skin Cancer Smartphone Applications for German-speaking Patients: Review and Content Analysis Using the Mobile App Rating Scale. Acta Derm Venereol 2019, 99, 1043–1044, doi:10.2340/00015555-3240.
  7. Kalhori, S.R.N.; Hemmat, M.; Noori, T.; Heydarian, S.; Katigari, M.R. Quality Evaluation of English Mobile Applications for Gestational Diabetes: App Review using Mobile Application Rating Scale (MARS). Curr Diabetes Rev 2020, doi:10.2174/1573399816666200703181438.
  8. Knitza, J.; Tascilar, K.; Messner, E.-M.; Meyer, M.; Vossen, D.; Pulla, A.; Bosch, P.; Kittler, J.; Kleyer, A.; Sewerin, P.; et al. German Mobile Apps in Rheumatology: Review and Analysis Using the Mobile Application Rating Scale (MARS). JMIR Mhealth Uhealth 2019, 7, e14991, doi:10.2196/14991.
  9. Rl, R.; F, K.; M, H.; A, O.; I, M.; S, H. Quality of Deaf and Hard-of-Hearing Mobile Apps: Evaluation Using the Mobile App Rating Scale (MARS) With Additional Criteria From a Content Expert Available online: https://pubmed.ncbi.nlm.nih.gov/31670695/ (accessed on Nov 30, 2020).
  10. Davalbhakta, S.; Advani, S.; Kumar, S.; Agarwal, V.; Bhoyar, S.; Fedirko, E.; Misra, D.; Goel, A.; Gupta, L.; Agarwal, V. A systematic review of the smartphone applications available for coronavirus disease 2019 (COVID19) and their assessment using the mobile app rating scale (MARS). medRxiv 2020, doi:10.1101/2020.07.02.20144964.
  11. Li, Y.; Zhao, Q.; Cross, W.M.; Chen, J.; Qin, C.; Sun, M. Assessing the quality of mobile applications targeting postpartum depression in China. Int J Ment Health Nurs 2020, 29, 772–785, doi:10.1111/inm.12713.
  12. Bardus, M.; van Beurden, S.B.; Smith, J.R.; Abraham, C. A review and content analysis of engagement, functionality, aesthetics, information quality, and change techniques in the most popular commercial apps for weight management. International Journal of Behavioral Nutrition and Physical Activity 2016, 13, 35, doi:10.1186/s12966-016-0359-9.
  13. Sharif, M.O.; Alkadhimi, A. Patient focused oral hygiene apps: an assessment of quality (using MARS) and knowledge content. Br Dent J 2019, 227, 383–386, doi:10.1038/s41415-019-0665-0.
  14. Choi, Y.K.; Demiris, G.; Lin, S.-Y.; Iribarren, S.J.; Landis, C.A.; Thompson, H.J.; McCurry, S.M.; Heitkemper, M.M.; Ward, T.M. Smartphone Applications to Support Sleep Self-Management: Review and Evaluation. J Clin Sleep Med 2018, 14, 1783–1790, doi:10.5664/jcsm.7396.
  15. Kim, B.Y.; Sharafoddini, A.; Tran, N.; Wen, E.Y.; Lee, J. Consumer Mobile Apps for Potential Drug-Drug Interaction Check: Systematic Review and Content Analysis Using the Mobile App Rating Scale (MARS). JMIR Mhealth Uhealth 2018, 6, e74, doi:10.2196/mhealth.8613.
  16. Riley, W.T.; Glasgow, R.E.; Etheredge, L.; Abernethy, A.P. Rapid, responsive, relevant (R3) research: a call for a rapid learning health research enterprise. Clin Transl Med 2013, 2, 10, doi:10.1186/2001-1326-2-10.

Reviewer 2 Report

This paper has numerous shortcomings, but one issue is the most important among other ones. Namely, what is the real added-value of this paper from the scientific point of view? In this paper Authors used the MARS scale to assess the quality 30 of the apps. The results of such as procedure can be important for the final users of these apps only. It has no scientific value to be published in a scientific journal. It would be a completely different situation if Authors would propose a modified or alternative version of the MARS scale and demonstrate its advantage over original one. In this case we could talk about some scientific input. Instead of this they used existing MARS scale and just made their calculations.

There are also other shortcomings of this paper and they include:

  • lack of research questions and hypotheses,
  • very general description of the MARS scale,
  • lack of calculations for the content presented in the table 3,
  • very brief and general conclusion section,
  • not very impressive number of references.

Author Response

Response to Reviewer 2 Comments

We thank the reviewer for their feedback. We provide a point-by-point answer to each specific comment. In addition, changes at the manuscript are provided with the track-changes mode.

Comments:

This paper has numerous shortcomings, but one issue is the most important among other ones. Namely, what is the real added-value of this paper from the scientific point of view? In this paper Authors used the MARS scale to assess the quality 30 of the apps. The results of such as procedure can be important for the final users of these apps only. It has no scientific value to be published in a scientific journal. It would be a completely different situation if Authors would propose a modified or alternative version of the MARS scale and demonstrate its advantage over original one. In this case we could talk about some scientific input. Instead of this they used existing MARS scale and just made their calculations.

Author’s response: Thank you for your feedback to improve our work.

We understand your concern about the scientific value of this study. The fact that the greatest beneficiary of the study will be the final users would be not a negative aspect. However, we think that practitioners will obtain valuable information to help them choose the best tool for their patients. Saving the distances, the value would be comparable to that provided by a scientific review about some health treatment, which is usually used by other researchers but especially by practitioners in order to base their therapeutic recommendations on a scientific basis.

There are currently hundreds of thousands of mHealth apps publicly available. Many of them are simply a catalogue of recommendations; some of them work as a follow-up tool, complementing an intervention program; while other mHealth apps are connected to some sensor, in order to offer interactive monitoring of user’s health. Unfortunately, these mHealth apps enter the mobile apps market with limited filters or controls that usually do not take into account aspects relevant in any health intervention, also in mHealth interventions [1].

MARS is a useful and reliable tool for evaluating mHealth apps and this type of scientific literature helps practitioners (and users) choose the best apps to recommend to their patients based on an exhaustive analysis of the characteristics and content. Otherwise, it would be difficult for practitioners, due to time issues, to carry out this type of exhaustive analysis for each of the pathologies they usually treat in their patients.

This has been understood in this way by many authors, reviewers, and editors, and there is an extensive literature with dozens of articles in the scientific literature evaluating the quality of mHealth apps using MARS in pain [2], behavior change [3], skin cancer [4], gestational diabetes [5], rheumatology [6], quality of deaf and hard-of-hearing [7], COVID-19 [8], postpartum depression [9], mindfulness [10], weigth management [11], oral hygiene apps [12], sleep self-management [13], or potential drug-drug interactions [14], among many others.

It is important to highlight that, according to its developers, the MARS was developed to provide researchers, professionals, and clinicians with a brief tool for classifying and assessing the quality of mHealth apps [15]. The MARS is an objective, reliable and easy-to-use tool, but it requires proper training of its evaluators and that they have knowledge about the specific health area. In this study, both reviewers were physical therapists with experience in low back pain management and mHealth field, previously trained in the use of MARS. There is also a version of MARS for app users (uMARS)[16], but in this study, the standard version for professionals has been used.

There are also other shortcomings of this paper and they include:

lack of research questions and hypotheses,

Author’s response:  Thank you for your comment. We have improved the proposition of the research question and the hypotheses. Similar to what has been done by other studies that have been carried out with this approach, the approach starts from the assumption that an exhaustive analysis will help users and practitioners to choose an appropriate mHealth for the health problem of low back pain. “It is suggested that an exhaustive analysis will allow practitioners and users to have relevant information for choosing an appropriate mHealth app. Thus, an exhaustive assessment of mobile apps available for interventions on LBP might be useful for clinicians, allowing them to know those most suited to encouraging patients with chronic LBP to become more active. The aim of this study is therefore to review which mobile apps are currently available in the app market, and to carry out an exhaustive assessment of them quality using the MARS, to provide an overview of their features, quality and practicability.”

very general description of the MARS scale,

Author’s response: Thanks for your feedback. The description of MARS has been expanded in all sections of the manuscript, improving the information available in the manuscript.

lack of calculations for the content presented in the table 3,

Author’s response: Thank you for your comment. We have noticed that the information describing the way in which the score obtained by both reviewers was used in each item was missing. The score of each reviewer in each item was used to obtain an average of both reviewers in each of the items and sections of the MARS, a value that was later used to calculate the mean of all the apps. “Each reviewer independently assessed each of the MARS items in each of the apps. For each item of each app, the mean value between the values of both reviewers was calculated. This mean value was used for the analysis of the mean of all the apps for each item and section of the MARS, obtaining a mean and a standard deviation.”

very brief and general conclusion section,

Author’s response: Thank you for your comment. The conclusion has been expanded with more precise information about the findings.

not very impressive number of references.

Author’s response: Thank you for your comment. The manuscript has been better referenced and, therefore, the number of references increased significantly (now 54 references).

References

  1. Boulos, M.N.K.; Brewer, A.C.; Karimkhani, C.; Buller, D.B.; Dellavalle, R.P. Mobile medical and health apps: state of the art, concerns, regulatory control and certification. Online J Public Health Inform 2014, 5, 229, doi:10.5210/ojphi.v5i3.4814.
  2. Salazar, A.; de Sola, H.; Failde, I.; Moral-Munoz, J.A. Measuring the Quality of Mobile Apps for the Management of Pain: Systematic Search and Evaluation Using the Mobile App Rating Scale. JMIR Mhealth Uhealth 2018, 6, doi:10.2196/10718.
  3. McKay, F.H.; Wright, A.; Shill, J.; Stephens, H.; Uccellini, M. Using Health and Well-Being Apps for Behavior Change: A Systematic Search and Rating of Apps. JMIR Mhealth Uhealth 2019, 7, e11926, doi:10.2196/11926.
  4. Steeb, T.; Wessely, A.; French, L.E.; Heppt, M.V.; Berking, C. Skin Cancer Smartphone Applications for German-speaking Patients: Review and Content Analysis Using the Mobile App Rating Scale. Acta Derm Venereol 2019, 99, 1043–1044, doi:10.2340/00015555-3240.
  5. Kalhori, S.R.N.; Hemmat, M.; Noori, T.; Heydarian, S.; Katigari, M.R. Quality Evaluation of English Mobile Applications for Gestational Diabetes: App Review using Mobile Application Rating Scale (MARS). Curr Diabetes Rev 2020, doi:10.2174/1573399816666200703181438.
  6. Knitza, J.; Tascilar, K.; Messner, E.-M.; Meyer, M.; Vossen, D.; Pulla, A.; Bosch, P.; Kittler, J.; Kleyer, A.; Sewerin, P.; et al. German Mobile Apps in Rheumatology: Review and Analysis Using the Mobile Application Rating Scale (MARS). JMIR Mhealth Uhealth 2019, 7, e14991, doi:10.2196/14991.
  7. Romero, R.; Kates, F.; Hart, M.; Ojeda, A.; Meirom, I.; Hardy, S. Quality of Deaf and Hard-of-Hearing Mobile Apps: Evaluation Using the Mobile App Rating Scale (MARS) With Additional Criteria From a Content Expert Available online: https://pubmed.ncbi.nlm.nih.gov/31670695/ (accessed on Nov 30, 2020).
  8. Davalbhakta, S.; Advani, S.; Kumar, S.; Agarwal, V.; Bhoyar, S.; Fedirko, E.; Misra, D.; Goel, A.; Gupta, L.; Agarwal, V. A systematic review of the smartphone applications available for coronavirus disease 2019 (COVID19) and their assessment using the mobile app rating scale (MARS). medRxiv 2020, doi:10.1101/2020.07.02.20144964.
  9. Li, Y.; Zhao, Q.; Cross, W.M.; Chen, J.; Qin, C.; Sun, M. Assessing the quality of mobile applications targeting postpartum depression in China. Int J Ment Health Nurs 2020, 29, 772–785, doi:10.1111/inm.12713.
  10. Mani, M.; Kavanagh, D.J.; Hides, L.; Stoyanov, S.R. Review and Evaluation of Mindfulness-Based iPhone Apps. JMIR Mhealth Uhealth 2015, 3, e82, doi:10.2196/mhealth.4328.
  11. Bardus, M.; van Beurden, S.B.; Smith, J.R.; Abraham, C. A review and content analysis of engagement, functionality, aesthetics, information quality, and change techniques in the most popular commercial apps for weight management. International Journal of Behavioral Nutrition and Physical Activity 2016, 13, 35, doi:10.1186/s12966-016-0359-9.
  12. Sharif, M.O.; Alkadhimi, A. Patient focused oral hygiene apps: an assessment of quality (using MARS) and knowledge content. Br Dent J 2019, 227, 383–386, doi:10.1038/s41415-019-0665-0.
  13. Choi, Y.K.; Demiris, G.; Lin, S.-Y.; Iribarren, S.J.; Landis, C.A.; Thompson, H.J.; McCurry, S.M.; Heitkemper, M.M.; Ward, T.M. Smartphone Applications to Support Sleep Self-Management: Review and Evaluation. J Clin Sleep Med 2018, 14, 1783–1790, doi:10.5664/jcsm.7396.
  14. Kim, B.Y.; Sharafoddini, A.; Tran, N.; Wen, E.Y.; Lee, J. Consumer Mobile Apps for Potential Drug-Drug Interaction Check: Systematic Review and Content Analysis Using the Mobile App Rating Scale (MARS). JMIR Mhealth Uhealth 2018, 6, e74, doi:10.2196/mhealth.8613.
  15. Stoyanov, S.R.; Hides, L.; Kavanagh, D.J.; Zelenko, O.; Tjondronegoro, D.; Mani, M. Mobile App Rating Scale: A New Tool for Assessing the Quality of Health Mobile Apps. JMIR mHealth and uHealth 2015, 3, e27, doi:10.2196/mhealth.3422.
  16. Stoyanov, S.R.; Hides, L.; Kavanagh, D.J.; Wilson, H. Development and Validation of the User Version of the Mobile Application Rating Scale (uMARS). JMIR Mhealth Uhealth 2016, 4, e72, doi:10.2196/mhealth.5849.

Reviewer 3 Report

Thank you for allowing me to review your submitted paper.

Overall I found the paper interesting. However, I believe it can be greatly enhanced with further development (supported by evidence) of your interpretation/discussion section.

I have made comments where improvements can be made of the attached PDF.

Author Response

Response to Reviewer 3 Comments

Comments:

Thank you for allowing me to review your submitted paper. Overall I found the paper interesting. However, I believe it can be greatly enhanced with further development (supported by evidence) of your interpretation/discussion section.

Author’s response: We thank you for your feedback. We provide a point-by-point answer to each specific comment. In addition, changes at the manuscript are provided with the track-changes mode.

I have made comments where improvements can be made of the attached PDF.

Author’s response: Thanks for your feedback. The proposed comments in the PDF have been resolved. In the case of the simplest ones (spelling errors, suggestion to change the term used, etc.) they have simply been solved by means of control of changes in Word. In the case of those that require a response, they are detailed below.

Why cost if you are excluding paid apps?

This needs to be clear.

Author’s response: The extraction form included the cost of the apps to be able to determine which of the apps were free and which of them had additional paid content included in the app. This has been explained in the manuscript.

Is it subjective or objective?

Author’s response: The MARS includes a subjective assessment that is scored independently of the content assessed in the objective sections. In this specific case, the mentioned section E refers to the subjective part of the MARS.

Can you explain what you mean by descriptive analysis?

Author’s response: With the term descriptive analysis we mean to obtain, organize, present and describe the data of the apps in order to facilitate the use, generally with the support of tables and numerical measures.

“tertile” is a statistical term used in a qualitative analysis

Author’s response: Although it is true that the term tertile is also used in qualitative research, its use in this study responds to the intention of dividing or classifying the valued apps in a distribution in several equal parts. In this way, and given the possible bias of presenting a classification based on the score, we think that its classification among the 33.3% of the best rated or worst rated apps may be useful for the reader. This classification by tertiles, also passed in the score, has biases but generally limits the evaluation of an application better or worse by having a few tenths more or less than another. This has been done in similar studies such as Salazar et al. (2018)[1]

  1. Salazar, A.; de Sola, H.; Failde, I.; Moral-Munoz, J.A. Measuring the Quality of Mobile Apps for the Management of Pain: Systematic Search and Evaluation Using the Mobile App Rating Scale. JMIR Mhealth Uhealth 2018, 6, doi:10.2196/10718.

App Store or iOS be consistent or explain.

Author’s response: The App Store is the place where you can find applications for the iOS operating system. An attempt has been made to clarify the terminology and verify a homogeneous use throughout the manuscript.

To this point, the way you have described you synthesis you still have 398 apps? This section need transparency and clarity

Author’s response: Thank you for your suggestion to clarify this part of the process. Once the duplicates were removed, the same independent reviewers screened the download page of the remaining results, and any apps that did not meet the selection criteria were deleted. In case of doubt, the apps were downloaded to verify compliance with the selection criteria. A third reviewer was consulted in the event of any doubts in relation to the eligibility of an app. In this way, only 34 Android apps and 15 iOS apps were selected as potentially eligible and were analyzed through the extraction form. This has been best described in the methods and results section.

Methods: “The same independent reviewers screened the title and the download page of the found apps. All potentially eligible apps were imported into a database (in case of doubt, the apps were also selected), and duplicates from apps found in both regions were identified and removed. Remaining apps were downloaded and any apps that did not meet the selection criteria were deleted. A third reviewer was consulted in the event of any doubts in relation to the eligibility of an app.“

Results: “A total of 500 apps were retrieved from the Play Store of the United Kingdom and Spain, while 53 apps were found in the App Store of both countries. After deleting duplicates and screening the title and the download page of the remaining apps, 34 Android and 15 iOS apps were selected as potentially eligible. After downloading and checking fulfilment of the selection criteria, 17 apps were finally included in the qualitative analysis..”

Line 160. This section needs to be more clearly written it is confusing what are you telling

Author’s response: Thank you for you suggestion. This section has been rewritten for ease of understanding and the part referenced in the methods section has been checked for consistency: “As described above in methods, the MARS divides the app scores into three dimensions. The mean quality score of the app is provided by the assessment of sections A-D . Subjective quality is scored according to the assessment of Section E. Finally, section F assesses the perceived impact of the app. This evaluation structure prioritises aspects such as engagement, functionality, aesthetics, and information over the subjective opinion of the assessor about the potential effectiveness of the app (E, subjective quality; F, perceived impact of the application).”

Which apps

Author’s response: The name of the apps that establish the minimum and maximum values in each of the aforementioned aspects has been included in the text of the manuscript to facilitate reading.

Line 185 (MARS assessment of technical criteria VS users’ rating in app platforms) Are you regarding this as an objective measure?

Author’s response: MARS has been described by its developers and other authors as an objective tool, although some of the MARS items are possibly an attempt to objectify subjective criteria.

Regarding the evaluation 1-5 (numerical or with stars) of the app platforms, which try to give a subjective score to the app in general, tools such as MARS ensure the evaluation of specific aspects of the apps independently, producing a deeper analysis of these aspects.

This text fragment has been modified to introduce this aspect: The MARS score is an objective tool based on technical criteria considered necessary for an appropriate application of the intervention. On the other hand, user assessment is usually subjective and, although it does not refer to specific items, it is mostly influenced by impact and subjective appreciation of the user.

Line 191 and 216(Digital tailored interventions: behavior change techniques; Privacy and confidentiality) These sections need to be related to the apps. Which better support your comments? which least support your comments?

Author’s response: Both sections have been better geared towards mHealth apps and numerous references have been included to support the arguments given.

“Theoretical strategies” What theory is this?

Author’s response: “Theoretical strategies” is the name that the MARS gives to a list of strategies that apps may be using to promote behavior change. In the same way, a list of technical aspects used to improve the scope of these strategies is suggested. This has been better explained in the manuscript: “MARS has a list of theoretical background/strategies that may have been applied in applications to promote behaviour change. These strategies are based on evaluation, feedback, information and monitoring, the choice of goals and the proposal of advice and training methods, among other behavior change techniques. Additionally, it has a list of possible technical aspects that enhance these strategies. The evaluator marks the strategies used by each app.”

Line 202 Can you cite it please?

Author’s response: This phrase refers to the apps included in the study. The sentence has been modified to clarify this aspect: “In the included apps, the most widely used technical aspect in order to favour these strategies was sending reminders.”

Citation style

Author’s response: The citations with errors have been corrected.

Line 222 Consider suggesting MFI security is now the recommended approach

Author’s response: We are not sure what you mean by the term MFI. For example, you can refer to Apple's MFi project to facilitate communication between its components. If you can clarify this point, we will be happy to delve into the subject.

Are you talking about 3rd party use? Be clear

Author’s response: This phrase refers to both the use of the information by the developers and a third party use. This information has been clarified: Storage and possible use of personal data by app developers or a third party use are other fundamental aspects to analyse in the preservation of privacy and confidentiality.”

Round 2

Reviewer 1 Report

No further comments are needed.

Reviewer 2 Report

I accept responses to my remarks and changes implemented by Authors.

Reviewer 3 Report

Thank you for considering and addressing the  comments offered on your original draft. I am happy to recommend accepting your paper in its current form.